# EMERGENT PASSWORD SIGNALLING IN THE GAME OF WEREWOLF

**Olaf Lipinski[1], Adam J. Sobey[1,2], Federico Cerutti[3] & Timothy J. Norman[1]**

[1]University of Southampton, [2]The Alan Turing Institute, [3]University of Brescia
{o.lipinski, ajs502, t.j.norman}@soton.ac.uk
federico.cerutti@unibs.it

## ABSTRACT

Emergent communication can lead to more efficient problem-solving heuristics and more domain specificity. It can perform better than a handcrafted communication protocol, potentially directing autonomous agents towards unforeseen yet effective solutions. Previous research has investigated a social deduction game, called Werewolf, where two groups of autonomous agents, villagers and werewolves, interact in an environment named RLupus. We study the impact of allowing the agents to communicate through multiple rounds and evaluate their language and performance against the baseline environment. We show that agents develop a highly successful heuristic using a single word vocabulary. They create an approach using passwords, allowing them to determine which agents are werewolves, which is the winning condition. We explore the possible reasons behind this strategy, with further experimental analysis showing that our approach speeds up the convergence of the agents towards a common communication strategy.

## 1 INTRODUCTION

Emergent communication allows agents to develop their own language as they interact. This allows for a range of benefits, such as a bespoke vocabulary or syntax, for a given environment and set of actions, enabling efficient communication and even optimising the agents' behaviour. The ability for the language to be completely adapted to the environment allows for unique and better strategies, as the whole meaning behind the communication channel can be adapted.

Different aspects of how the communication emerges, and its properties, have been investigated, with one of the first works examining them in the setting of deep learning being that of Lazaridou et al. (2017), with previous works using other techniques such as iterated learning Smith et al. (2003) or evolutionary algorithms (Steels, 2003). Most of the emergent communication literature focuses on relatively simple non-situated settings, with the most common being referential games, where two agents have to describe and identify a picture from a description (Lazaridou et al., 2017; 2018; Luna et al., 2020; Dessì et al., 2021). In such settings, there have also been some attempts to consider multiround communication (Evtimova et al., 2018; Harding Graesser et al., 2019; Kottur et al., 2017), with promising results in terms of the performance and generalisation of the agents' language. However, the work of Kottur et al. (2017) shows that while the language may be effective at solving the tasks that the agents are presented with, it is not necessarily interpretable (i.e., understandable by humans) nor compositional (i.e., consisting of symbols that can be combined to either create a new meaning or convey already learned information).

Emergent communication has also been explored as a way to improve the performance of intelligent agents in the game of Werewolf (Brandizzi et al., 2021). In this game, players are divided into two teams, villagers and werewolves, where they have to use their social and deduction skills to identify players from the opposing team. The performance of the villagers significantly improves when the agents are allowed to create their own language, instead of using a prescribed protocol, however the analysis of the language and strategy of the villagers is left for future work. In addition, in the original environment as presented by (Brandizzi et al., 2021), the villagers only had a single round

of communication to agree on a player to vote out as an alleged werewolf, which does not allow them to fully explore other strategies.

In this paper, we determine the effect of a longer period of discussion on the language developed and the success rate in identifying the werewolves by introducing multiple rounds of communication. Our work is motivated by the assumption that if agents are allowed more time to communicate, then they will develop more successful strategies, when compared to handcrafted solutions. We have seen such improvements already, as shown by Brandizzi et al. (2021) in the AI Wolf competition Toriumi et al. (2017), but only for a single round of communication. We introduce multiple rounds instead of just allowing the agents to play for longer, as multiple rounds allow for the possibility of dialogue emergence, which would be impossible with a single communication round.

## 2 GAME OF WEREWOLF

The game of Werewolf begins with the assignment of roles to the players. Some players are assigned the role of werewolves, while others are villagers. To balance the game, the ratio of the roles is usually at most one werewolf to three villagers.

The game has two phases in which different actions can be taken, "daytime" and "nighttime". The first phase is "nighttime", where the werewolves are allowed to discuss and choose a villager to vote out. The discussion and the votes of the werewolves are only visible to the werewolves. After a player has been chosen by the werewolves, they are removed from the game, and the game progresses to the "daytime" phase. The "daytime" phase consists of one or more rounds of communication, where players are allowed to exchange a single message per communication round. During this phase, players are meant to discuss the events that have occurred during all previous phases. Then, all players have to vote to eliminate someone that they think is a werewolf. After the "execution", the game loops back to the "nighttime" phase, and the phases alternate until either team wins the game.

The goal of each team is to eliminate the opposing team. The werewolves can eliminate villagers during both the "nighttime", by voting them out, and the "daytime", by misleading the villagers to vote for each other. Villagers in turn only have the "daytime" phase to work out who the werewolves are and vote them out. The game ends when all werewolves are voted out, or when the number of villagers and werewolves are equal. This condition is introduced because when the numbers are equal on each team, the villagers cannot win any more, as all votes would be ties.

We provide a visual representation of the game phases in Appendix A and Figure 4. For further details, including a theoretical analysis of the win rates when played with and without emergent communication, we refer to the detailed analysis by Brandizzi et al. (2021).

### 2.1 AGENT ENVIRONMENT

We will be basing our reinforcement learning environment on that of Brandizzi et al. (2021). This environment uses the Ray (Moritz et al., 2018) and RLLib (Liang et al., 2018) libraries to train all the villagers. For training, we use the APPO algorithm, which is an asynchronous sampling variant of the Proximal Policy Optimization (Schulman et al., 2017), provided through RLLib (Liang et al., 2018). Our agents are Ray LSTM wrapped (Liang et al., 2018) parametric agents with linear layers, which process the observation data and allow for the message generation. The input for the agents is an observation from the environment itself, written with OpenAI Gym (Brockman et al., 2016), and the output is an action to take in the environment, in our case the message and vote of a given agent, for the current round of communication. We provide further information regarding the technical details of our environment and agents in Appendix B

We extend the Werewolf environment to a multi-round communication domain and introduce changes to how the voting works to incentivise quicker convergence on communication between the villagers. [1]

The multi-round approach is implemented during the "daytime" phase when all agents, including the werewolves, are permitted to communicate. This is done as the werewolves in the original envi-

---

[1]Our code is available on GitHub at https://github.com/olipinski/rl_werewolf

ronment have a static policy[2] that they have to use (Brandizzi et al., 2021), meaning communication during the "nighttime" phase has no effect on the werewolves' vote. Our game can be parameterised with the number of communication rounds that the agents are allowed to have, therefore allowing us to vary the amount of time agents have to converse, and measure the effect of this variation.

Secondly, we also incentivise our agents to decide quickly on the target of their vote, while also maintaining a high consensus rate among them as to who their target will be. This is done through the agreement loss, as presented by Brandizzi et al. (2021), as well as an indirect loss based on the number of rounds taken to reach a conclusion. The agreement loss penalises agents who voted for a target that was not voted out, while our voting threshold is implemented through a required percentage of votes for the voting outcome to be valid. This means that, when less than X% of agents agree on a target, then the vote is considered invalid, and no players are eliminated. This modification can be viewed as adding an independent judge to count all agent votes, where only a significant plurality is permitted to decide whether to vote out an agent or not. This contrasts with the original environment, which only required a simple plurality of agents to choose a target, even if that plurality amounted to just two agents.

## 3 EVALUATION OF LANGUAGE & PARAMETERS

Our modified version of the environment exposes two additional parameters to explore: the number of rounds and the voting threshold. We explore them using a grid search, maintaining the other parameters from the original paper (Brandizzi et al., 2021). Further technical details are available in Appendix B.

As shown by Brandizzi et al. (2021), their agents have already improved upon the calculated theoretical baseline win rates for random policies, through allowing them to use emergent protocols within a single round of communication. Our agent and game configuration show improvements over both the theoretical baseline win rates and the previously demonstrated performance (Brandizzi et al., 2021), as we show in Table 1. We define the win rate as the percentage of games that the villagers win in a single training run, consisting of multiple rotations between "daytime" and "nighttime" phases until one team wins. We further note that for certain configurations of the round count and threshold, our agents achieve a 100% win rate. We consider that our comparison to the original work is adequate, as the core rules of the game have not been changed. We only change the communicative part of the game and add the reward to match, with otherwise the agents and the environment staying the same.

### 3.1 PASSWORD SIGNALLING STRATEGY

During the game, villagers are observed to send the same message every round of communication and vote off those who do not comply with this strategy. As werewolves do not learn because they have a static policy, this strategy allows villagers to easily distinguish the alternative agent type. For this tactic, multiple rounds of communication help the agents develop this test faster, and do not require a random vote at the beginning. For a single communication round, the agents at the start of the game have no information about whom the werewolves may be, and so would need to vote for a random player. Instead, with multiple rounds, they can establish this distinction without having to cast the final vote. We consider this strategy to be of particular interest, as we believe it resembles that of the Turing Test (Turing, 1950), with it being performed by multiple separate intelligent agents on each other, with no involvement of humans.

Analysing the language that our agents have developed for this strategy, we find a focus on a sparse word vocabulary for the successful tactics, while most unsuccessful strategies have multiple words in their vocabulary. Moreover, almost all successful agent populations use their single word at least 90% of the time, with a minor number of outliers. Therefore, our agents do not develop a compositional language, but rather a kind of password. We speculate that this is because any additional words would not bring an improvement to their performance. With just this code, they can already achieve a high, if not the maximal, win rate, and therefore reward. Hence, we could say that this strategy creates a very efficient language, for this specific environment, where the werewolves use a static policy.

---

[2]We explain the details of the static policy in Appendix B

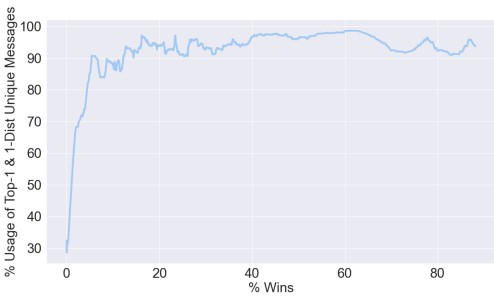

Figure 1: Most used unique message and its top ten distance-one adjacents increasing as the villager win percentage increases. This positive correlation shows winning strategies mostly use a single unique message, with some exploration of adjacent messages (i.e. using (1,1,1,0) instead of previously used (1,1,0,0))

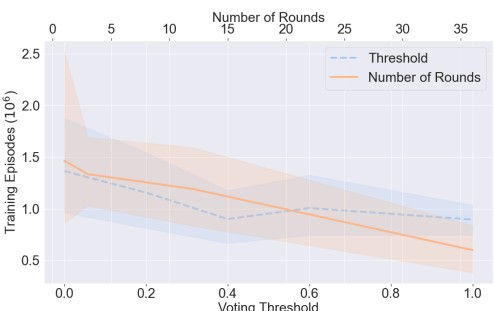

Figure 2: Impact of both the voting plurality threshold and number of rounds on the convergence speed. As both parameters increase, we can observe the decrease in the average number of episodes required to reach the 75% win rate. Shaded areas represent the 95% confidence interval.

We can see the usage of the most common word, and its distance-one adjacents, defined as any message that has a single character difference from the most common message, in Figure 1. We can observe the same correlation without including the distance-one adjacents, however their inclusion smooths out the graph. We gather the word usage information from the latter parts of the training of our agents, focusing on learners where the strategy has already developed and has completed its exploration phase. We consider this approach to be the same as freezing the agents' policies, as the win rate improvements plateau during the second half of the training time and there is minimal variation in the policies at this stage.

## 3.2 CONVERGENCE SPEED

We have observed that our modifications impact the convergence speed significantly. We define the convergence point as the episode number where our agents reach over 75% win rate, which we chose as an arbitrary threshold for a successful strategy. Both the number of rounds and the voting threshold decrease the average convergence episode. We find that the higher the number of communication rounds, the quicker the agents converge. Furthermore, we can see that the average convergence speed increases as the voting threshold increases. Both of these relationships are visualised in Figure 2. This means that enforcing a higher agreement rate between agents could also positively impact the speed with which they develop a common language or code. The number of rounds also positively affects the convergence and decreases the amount of time that is required to find a good strategy, observed in Figure 2.

Statistical analysis is performed on the impact that both the number of rounds and the voting threshold have on the convergence speed. Our results show that the number of rounds has a statistically significant effect on both win rate and convergence speed. This confirms that the more time the agents have to converse, the quicker they can converge on a common strategy. The voting threshold, however, does not have a statistically significant effect. [3] Our inclusion of the voting threshold was supposed to incentivise faster convergence in the agents, by penalising agents that vote for different agents each round. The negative result for the voting threshold may be explained by the voting threshold being too close in function to the agreement reward, as implemented by Brandizzi et al. (2021).

## 3.3 COMPARISON TO THE ORIGINAL ENVIRONMENT

The authors of RLupus (Brandizzi et al., 2021) present interesting findings in terms of how emergent communication affects performance in the game of Werewolf. They show that, with certain configurations for the message length and vocabulary size, the agents which are allowed to create their own language will outperform others. Moreover, when compared with the AI Wolf competition (Toriumi et al., 2017), the emergent communication agents surpass their handwritten counterparts.

---

[3]For details of our statistical analysis, please see Appendix C.

Table 1: Results for both our and the original (Brandizzi et al., 2021) environment. Our configuration notation is consistent with Brandizzi et al. (2021), where "SR" is signal range (i.e. the number of characters available, where 2 would be binary), "SL" is signal length, and "PL" is the number of players. "WR" is assigned to "Win Rate", with "TH" and "RS" being threshold value and number of rounds respectively. Lastly, the "Convergence" column refers to the episode number, or convergence point, that the agents with that configuration achieved. All values are reported for the best run of each configuration, with better ones (as compared between the original environment and our modified version) displayed in **bold**. Mean $\pm$ standard deviation and t-Based 95% confidence interval are reported in the parentheses. Convergence episode for the original configurations is reported as obtained by our reproduction.

| Configuration | TH | RS | Our WR (%) | Original WR (%) | Convergence ($10^6$) |
|---|---|---|---|---|---|
| SL9-SR2-PL9 | 1 | 36 | **100** $(60 \pm 31, (32, 88))$ | N/A | **0.70** $(0.94 \pm 0.84, (-0.61, 2.49))$ |
| SL9-SR2-PL9 | 0 | 1 | 99 $(78 \pm 32, (40, 115))$ | 45 | 1.29 $(1.57 \pm 0.41, (0.81, 2.33))$ |
| SL9-SR2-PL21 | 1 | 3 | 96 $(83 \pm 13, (76, 89))$ | N/A | **0.70** $(1.47 \pm 0.42, (1.24, 1.7))$ |
| SL9-SR2-PL21 | 0 | 1 | 95 $(78 \pm 27, (60, 97))$ | **98** | 1.19 $(1.28 \pm 0.59, (0.86, 1.69))$ |
| SL21-SR2-PL21 | 0.4 | 3 | **100** $(80 \pm 17, (63, 96))$ | N/A | 0.62 $(0.93 \pm 0.19, (0.71, 1.14))$ |
| SL21-SR2-PL21 | 0 | 1 | 98 $(92 \pm 13, (80, 104))$ | 94 | **0.58** $(0.72 \pm 0.12, (0.6, 0.83))$ |

In Table 1 we compare our results to the original environment. With our modifications, we achieve a lower total episode count before convergence for two out of three compared configurations, while also achieving a higher win rate for two out of the three configurations. This shows that allowing the agents to communicate for longer offers to improve both metrics of the game. We have also included reproductions of the configurations that were presented by Brandizzi et al. (2021) with our modifications to the code for a better comparison. We can note that our reproductions have higher win rates than reported, which could mean that the original runs were not fully converged, possibly owing to the time or compute power available for the original study. Nevertheless, our results still improve over the reproductions.

## 4    CONCLUSIONS

Emergent communication is an area of research that is currently having a resurgence of interest, due to the success of techniques from deep learning. The aim is to allow agents to develop their own communication, that is optimised for the setting they operate in. The benefits are efficiency, reduced designer effort and creativity in problem-solving. We have introduced multi-round communication to the originally single round environment of Werewolf, as presented by Brandizzi et al. (2021), to study these properties. We show that the number of communication rounds decreases the convergence time of the agents, with statistical analysis showing that this correlation is significant. Finally, we investigate the strategies that the agents develop to achieve the high win rates, and show that our agents are using password signalling allowing villages to efficiently identity each other. Our results confirm that allowing agents to communicate for longer offers improvements to the main metrics of the environment, and may point towards new paths to explore for other settings.

ACKNOWLEDGMENTS

This work was supported by the UK Research and Innovation Centre for Doctoral Training in Machine Intelligence for Nano-electronic Devices and Systems [EP/S024298/1]

The authors would like to thank Lloyd's Register Foundation for their support.

The authors acknowledge the use of the IRIDIS High-Performance Computing Facility, and associated support services at the University of Southampton, in the completion of this work.

We would like to thank the authors of Brandizzi et al. (2021) for their helpful comments and discussion about using their environment.

We would also like to thank the anonymous reviewers for their helpful comments on improving our work.

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

## A    APPENDIX: FURTHER ANALYSIS OF THE WIN RATE

We can look at the direct impact that both of our additional parameters have on the average win rate of the villagers. Figure 3 illustrates the interaction between the parameters in terms of the win rate, with the shaded areas representing the 95% confidence interval.

When looking purely at the effect of the number of rounds, for a larger amount the win rates tend to be lower, as we can see in Figure 5. However, the number of rounds together with the voting threshold increases the average win rate past the previously reported values (Brandizzi et al., 2021), and results in an overall positive trend.

We hypothesise that this decrease is due to the longer training times, as the larger number of rounds requires more computation, and therefore does not achieve convergence within the same time as our other configurations. We can see that the number of total episodes decreases as the number of rounds increases, which is due to the limited amount of time that we could run the simulations in Figure 6.

## B    APPENDIX: TECHNICAL DETAILS

### B.1    HYPERPARAMETERS & TRAINING

For our parameter grid, we have used the values of [0, 0.2, 0.4, 0.6, 1] and [1, 3, 12, 36] for the threshold and number of rounds respectively. We run our simulations for an average of 3M episodes. With our experimental setup as described, we have performed a total of 180 runs for a total of 3600 hrs of GPU time. An average run of 3M episodes took ~20 hours on a single NVIDIA RTX8000 GPU.

### B.2    ENVIRONMENT AND AGENTS

Our agents use the APPO, or Asynchronous Proximal Policy Optimisation from RLLib (Liang et al., 2018), for learning. This algorithm is in turn based on the PPO algorithm (Schulman et al., 2017).

The reward scheme for the villagers is relatively simple, and seeks to incentivise specific behaviours. The rewards that we assign are -5 for death, as to tell our agents to avoid being voted out/eaten by the werewolf; +25 for winning the game, to incentivise winning strategies; -25 for losing the game so that the agents avoid any losing strategies; -1 for picking a target other than the one that was voted out, to reinforce uniform voting; and finally -2 for wasting a round, or not reaching the voting threshold, to further incentivise voting in unison.

The observations the agents receive are an OpenAI Gym (Brockman et al., 2016) Box type, which contains the messages of the other agents and their votes. The action that each agent can take is to produce a message and a vote for each round of communication. No other actions can be taken by the agents. After a successful vote, be it by the werewolves during the "nighttime", or all players in the "daytime" phase, the voted out player is removed from the game, and may take no further actions.

Table 2: Linear Regression Analysis. The win rate and convergence episode are very likely to be affected by the number of rounds. However, similarly to the previous tests, the voting threshold may not have an effect on either.

| Relationship | $p$-Value | $R^2$ |
|---|---|---|
| Number of Rounds vs Win Rate | $< 0.001$ | 0.264 |
| Voting Threshold vs Win Rate | 0.600 | 0.002 |
| Number of Rounds vs Convergence Episode | $< 0.001$ | 0.189 |
| Voting Threshold vs Convergence Episode | 0.686 | 0.001 |

Table 3: Normality Test Results. Our data is **not** normally distributed for both the win rate and the episode count.

| Variable | Test Type | $p$-Value | Statistic |
|---|---|---|---|
| Win Rate | Shapiro-Wilk | 0.000 | 0.858 |
| Win Rate | D'Agostino and Pearson | 0.000 | 2554.462 |
| Episode Count | Shapiro-Wilk | 0.000 | 0.659 |
| Episode Count | D'Agostino and Pearson | 0.000 | 67.593 |

The messages themselves are an array of integers. These can be selected by the agents from the available range called Signal Range (SR). The length of this array is in turn determined by the parameter Signal Length (SL).

The messages are passed during all phases as an observation to the agents. Depending on the agent's role, it can receive both the "nighttime" and "daytime" observations, if it is a werewolf, or purely the "daytime" observations, if it is a villager.

For our static werewolf policy, we follow the work done by Brandizzi et al. (2021), and use their static random target werewolf policy. This policy picks an agent at random to be voted out by the werewolves. All of our werewolves follow this static policy.

## C    APPENDIX: STATISTICAL SIGNIFICANCE

To verify our results, we have performed a statistical significance analysis. This analysis relies on the 180 data points that we have obtained through our simulations. However, as the number of data points per configuration are relatively low, the overall results for the regression analysis may not be fully accurate.

Firstly, we analysed the normality of the distribution of both of our dependent variables - the mean of villager win rates, and the number of episodes that it takes villagers to converge. We performed the normality analysis with two tests from the SciPy (Virtanen et al., 2020) package. We report the results of the tests in Table 3.

Secondly, we analysed the correlation and significance of the correlation with the Spearman correlation metric, as our data was not normally distributed. We used the function provided by the pandas (McKinney, 2010; pandas Development Team, 2020) package. We can see the results of these tests in Figure 7.

Finally, we analysed our results with a simple linear regression model, from the SciPy (Virtanen et al., 2020) package. We present the results of this analysis in Table 2.

Overall, from our analyses we can infer that the number of rounds does have a statistically significant effect on both the win rate and convergence speed, as we have theorised in the main body of this work. However, our analysis fails to find a relationship between the voting threshold and either of our dependent variables.

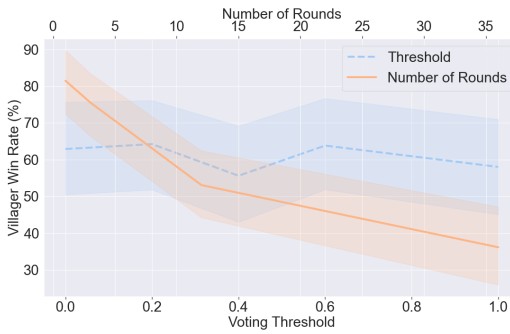

Figure 3: Impact of both the voting plurality threshold and number of rounds on the win rate. While the number of rounds decreases the average win rate, the threshold's impact is less negative.

Figure 4: Visual representation of the flow of the game of Werewolf. The game starts at the night phase, progresses to the day phase, and then loops back around to the night phase. The discussion in the night phase only includes werewolves, while in the day phase all players may converse.

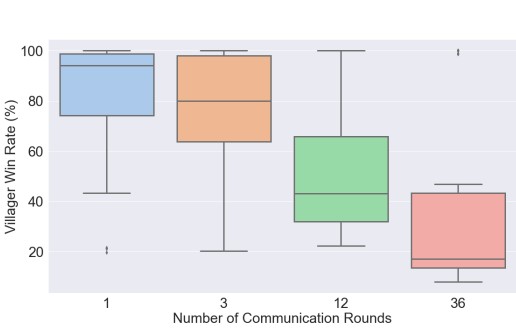

Figure 5: Impact of the number of communication rounds on the win rate of villagers. We can see that the number of communication rounds is negatively correlated with the average win rate.

Figure 6: Impact of the number of communication rounds on the number of training episodes. We hypothesise that this is the reason behind lower performance as the number of rounds increases. Due to a bigger simulations size, as the agents need to exchange more messages, the configurations with higher number of rounds perform worse, as they do not have enough time to converge.

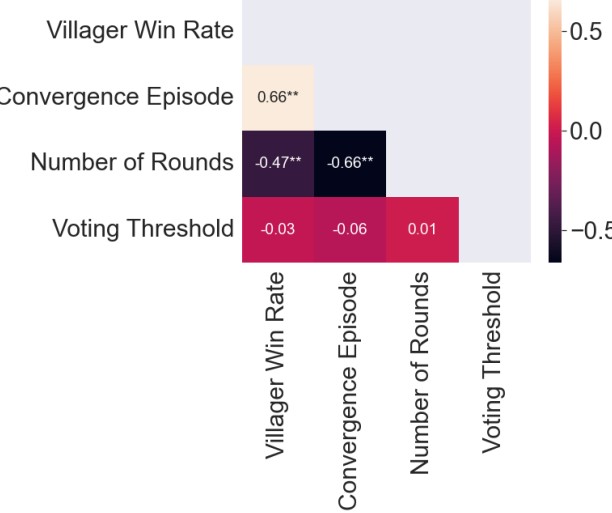

Figure 7: Spearman correlation strength and its significance. The strength for the number of rounds affecting either the villager win rate or convergence episode is high. However, the relationship between the threshold and win rate or convergence is much weaker. The significance can be discerned by the number of **\*** next to the corresponding number, where **no** \* signifies $p > 0.05$; \* is $p < 0.05$; and \*\* is $p < 0.01$

