# OpenReview forum: "Emergent Password Signalling in the Game of Werewolf"
_ICLR.cc/2022/Workshop/EmeCom — EmeCom Workshop at ICLR 2022_

### Official Review · Reviewer_RiJz · 2022-03-10
**Interesting results but weak form**

**Rating:** Weak accept
**Confidence:** 4

**Review:**

## Summary
The authors build on top of the RLupus environment, augmenting it with multi-round communication and voting threshold.
They report a positive trend between the number of rounds and the convergence speed (defined as the episode where agents reach over 75% winning rate [WR]) and a positive positive trend between the voting threshold and convergence speed. Moreover, they perform a statistical analysis that backs up the former trend but not the latter.
The authors hypothesize how the emerged language is more akin to a Turing test rather than showing natural language properties such as compositionality. They back up this claim with Figure 2.

## Findings and results

### Turing test
This paper provides insights on the problem of natural language proprieties not arising in emergent communication [EmeCom] frameworks. While I find interesting the comparison to the Turing test, I believe It can be misleading since it exaggerates the agents' planning skills. While the authors do not specify what kind of static policy is adopted by the werewolves [ww], the agents' ability to spot a random output is to be expected (given the loss definitions) and should not be compared to a deliberate intent to distinguish between ww and villagers [vil].
However, I find this an excellent discussion element and a starting point for future studies investigating adaptive ww agents.

### Convergence speed and multi-round
The authors also report a positive trend between the number of rounds and the convergence speed. I find it in line with the current SOTA and appreciate how the trend is also followed by short speculation on the motives.

### Convergence speed and voting threshold
On the other hand, I find this other trend quite confusing.
Indeed, the authors do not justify or speculate on the possible origins of this correlation, and they report how their statistical analysis does not yield significant results. While I do appreciate the transparency of their analysis, I question why the authors decide to include this information and if it should be included at all.

## Form and clarity
While reading the paper, I had various issues understanding what the authors were communicating.
There are several sentences that may require rewriting, especially regarding the explanation of Figure 1  and section 2.1.
Finally, the appendix contains 4 pages of additional material which are not included in the main body. I am aware of the 5 page limit for this submission, but such a long appendix looks to me like a way to bypass this limit. Especially considering how some sections are not relevant for the scope of the paper (i.e.  A.1, B.2) compared to the others.


## Conclusion
Overall, I believe these results could spark interesting discussions on the purpose of multi-round communication in the EmeCom field. Since this is the scope of the submission I believe this work can pass, but I strongly advise the authors to consider the reviews mentioned in the previous sections.

---

### Official Review · Reviewer_2zLR · 2022-03-22

**Rating:** Weak accept
**Confidence:** 4

**Review:**

### Summary:
The paper studies emergent multi-round communication in the context of a competitive social game (Werewolf). Prior work has studied single-round communication in this domain and has demonstrated that learned communication is superior to handcrafted analytical policies. This work extends prior work by considering multi-round communication, in which agents may exchange multiple messages before taking a vote on which agent to eliminate from the game. Empirical results in the RLupus environment show that multi-round communication outperforms single-round learned communication and analytical communication policies in both win rate and convergence speed.

### Strengths:
- The motivation and theme of the paper are well-suited to the theme of the workshop.
- Multi-round communication is an interesting/important extension to the Werewolf game. When played amongst humans, multi-round communication is key for achieving consensus before a vote (and gives the werewolves an opportunity to deceive the villagers).
- The paper does a good job at describing the facets of coordination / importance of efficient communication strategies to achieve success in this setting and shows improvements over prior work.

### Questions/Suggestions:
- The paper relies heavily on references to prior work for background information -- specifically Brandizzi et al. (2021) -- and would benefit from including relevant information directly. For example:
    - Section 2.2: What are the loss functions used to train villager policies? They are just mentioned in passing as being the same as the prior work. It would be better to define these objectives clearly somewhere in this paper.
    - Section 2.2: Additionally, what are the observations/actions of each agent? What are the messages? How are they structured? How are messages passed back and forth amongst villagers?
- Have the authors considered scenarios where the werewolves also learn to communicate / participate in voting?
    - It seems that multi-round communication would be more interesting in that setting. If werewolves can also communicate, the simple password / membership signal strategy of the villagers would be less effective (werewolves could learn this signal too). It would also enable much richer interactions, including deception by the werewolves.
- Section 3.1: I am not sure that "Turing test" is an accurate description of the strategy learned by the villagers. The villager strategy is more like a single-token membership signal / passcode, whereas the Turing test is a commentary on general intelligence as demonstrated through diverse conversation.
- Section 3.1: "We gather this information from the later parts of the training of our agents..."
    - Is this equivalent to freezing the policies after some number of training iterations and performing analyses at test-time? If not, this sort of analysis should be done at test-time across multiple random seeds.
- Table 1: Are these results reported for a single run? The results would benefit from the inclusion of mean/std-dev/confidence intervals across multiple runs / random seeds.
- Section 3.2: "Our results show that the number of rounds has a statistically significant effect on both win rate and convergence speed. This further confirms that the more time the agents have to converse..."
    - I am not sure that this is a reflection of "better" communication, but rather that increasing the # of rounds increases the likelihood that werewolves policies make a mistake (i.e. # of round makes the game easier rather than making communication richer or more effective).

---

### Decision · Program_Chairs · 2022-03-25

**Decision:**

Accept

**Comment:**

Both reviewers found the topic and approach interesting and would like to see it discussed further at the workshop. We hope the authors gain valuable feedback from the reviews and continue to iterate on their work